# Teaching and Researching in the Context of COVID-19: An Empirical Study in Higher Education

**Margarida Rodrigues** [1] , **Rui Silva** [2] **and Mário Franco** [1,*]

1 CEFAGE-UBI Research Center, Department of Management and Economics, University of Beira Interior, 6200-209 Covilhã, Portugal; mmmrodrigues@sapo.pt
2 CETRAD Research Center, University of Trás-os-Montes and Alto Douro—UTAD, 5000-801 Vila Real, Portugal; ruisilva@utad.pt
* Correspondence: mfranco@ubi.pt

**Abstract:** There is increasing recognition worldwide of the importance of academic activities, specifically in situations of pandemics. Therefore, this study aimed to understand the effects of COVID-19 on lecturers/researchers and Ph.D. and master students who have faced unexpected and continuous disruption in their teaching and research activities. To fulfil the aims, the study focused on a mixed method approach quantitative study based on a questionnaire administered on social networks and open questions. The unit of analysis was lecturers/researchers and Ph.D. and master students. The results obtained show that this lengthy interruption had severe impacts on their activities, requiring new competencies and capacities to deal with changes in a short period of time, including less positive feelings affecting them and their families. The main contribution of this study lies in identifying the barriers and opportunities created by this virus in the academic world and in presenting a theoretical framework to improve the situation, given that the confinement exponentiated negative and psychological feelings in academics, although telework is seen as a positive factor with continuity in the future, as a way to foster the social, environmental sustainability of Higher Education Institutions (HEIs) and the wellbeing of their human capital. As implications for practice, the evidence points to the need for academics to be provided with training in E-learning, about technological tools for use in distance-learning and to reconsider how they carry out their research on the ground.

**Keywords:** COVID-19; teaching; academic disruption; barriers and opportunities; HEIs

## 1. Introduction

The first cases of COVID-19 appeared in December 2019 in Wuhan, China, a city with around 11 million inhabitants, with this being reported to the World Health Organisation (WHO) on 31 December. This report stated that around 40 people were infected with atypical pneumonia caused by an unknown virus. However, this virus spread exponentially across borders, and by 11 March 2020, there were 118.000 cases of infection in 114 countries and 4.291 deaths, causing the WHO to declare the state of a pandemic [1]. Following this declaration, exceptional measures of lockdown and social distancing were adopted worldwide, with higher education institutions (HEIs) and their research centres being no exception [2]. This decision is reported in the vast literature on the closure of HEIs [3] in order to reduce the spread of infectious diseases and break transmission chains [4–6]. In this scenario of a growing pandemic, at the beginning of March, HEIs in Portugal suspended all their face-to-face activities and immediately changed to the online learning format [7]. This means that the "pandemic is likely to generate the greatest disruption in educational opportunity worldwide in a generation" [8] (p. 4).

This disruption meant that in parallel to matters of public health caused by the spread of COVID-19, other new issues arose in people's personal and professional lives [9], and this includes higher education lecturers, researchers and students [10]. This implied it was necessary to re-learn how to operate, function and communicate [11]. However,

Corbera et al. [11] argued that this should not be understood as a threat but rather as an opportunity to redefine excellence in teaching and research, to present even more respected and sustainable academic practices. From another perspective, Myers et al. [12] argued that researchers devoted less time to research activities, with the number of hours being heterogeneous as this varies according to areas of research and gender. In addition, the scarce literature on teaching and research during the COVID-19 pandemic confirms that resilience is a condition *sine qua non to* combat this virus, particularly by academics [13,14], that coronavirus provokes reactions of anxiety, stress [15], pessimism [16] and has adverse effects on research [17,18].

Furthermore, publications on this topic are found to be descriptive and exploratory, with short texts and little discussion, and minimal bibliographical references given the completely unknown nature of the disease and a great amount of non-peer-reviewed literature. In these circumstances, there is an urgent need for empirical, peer-reviewed studies with scientific validity in relation to an unknown topic so that scientific knowledge can evolve and prevent stagnation [19–25] and also begin to understand its extent [26]. With this connection, the literature reviewed on the topic analysed here, although recent and in irregular format, revealed some gaps. Dohaney et al. [13] identified a lack of research on how all academic actors respond, in practice, to the challenges presented by COVID-19; Corbera et al. [11] concluded that research was mostly in the area of health, and so it is urgent to investigate how coronavirus has affected researchers; Myers et al. [12] stated there was a lack of empirical evidence on the nature and magnitude of the impact of COVID-19 on scientific research.

Considering these gaps and intending to contribute to developing scientific knowledge, at a time requiring transversal [27] and multidisciplinary [28] research on this pandemic, this study aims to map and determine the main practical effects on HEI lecturers and researchers, using a mixed methodology to provide a response with internal and external validity. The main contribution of this article lies in its originality, i.e., the fact of discussing its results in the light of the literature, as many publications are mere synopses of events. In addition, the results obtained provide some implications for theory and practice, limitations and future research.

This brief introduction is followed by the literature review, methodology, results and conclusions.

## 2. Literature Review

### 2.1. Pandemics and Other Disasters in the HEIs

In the face of the global COVID-19 lockdown, HEIs' educational policies included strategies to ensure the continuation of the teaching system, involving various aspects such as the availability of virtual platforms, digital content and possible lecturer training, principally in developing countries [29]. Pandemics and other disasters place HEIs at risk, causing disruption in learning and teaching over a lengthy and undefined period, and so they had to develop and implement contingency plans to continue with their mission [13]. These plans must provide a response at various levels, such as health, research and development, and ensuring the continuity of teaching and access to platforms of bibliographic and technological resources, without neglecting the emotional level [29]. The enforced adaptation due to COVID-19 had significant educational, social and economic impacts on the lives of everyone, creating negative feelings and challenges for all, and teaching institutions' rapid response did not end with ensuring the continuity of learning [30].

However, this HEI provision leads to the question of the effects on the actors involved, with Strielkowski [31] arguing that teaching in the COVID-19 context includes resilience and institutional and technological support. Corbera et al. [11] stated that lecturers and researchers need to reorganise priorities, now and in the future, giving prominence to collective rather than individual objectives, maintaining their responsibilities to HEIs and being scientifically responsive. This means that during the pandemic, resilience has a central role [14], being understood as a phenomenon through which many people have

retained their mental health, despite exposure to serious events [32]. The importance of resilience is based on experiences reported by individuals in fighting other infectious diseases (e.g., Ebola, flu, H1N1, SARS), highlighting that social isolation, fear and frustration are predictors of psychological problems [33], which after quarantine are transformed into stress and anxiety [34–36]. Recently, Zanon et al. [14] concluded that these symptoms could be tackled through resilience, which can be stimulated through optimism, wellbeing and creativity, among other factors. However, the meaning of resilience varies according to the context in which it is applied and the disciplinary approach, as this concept has multiple meanings [13]. Therefore, when speaking about education, these authors argue that educational resilience must be separated from institutional resilience. Educational resilience is students' capacity to deal with and progress in the face of adversity [37]; it is a "state, a condition and a practice" [38] (p. 543). Institutional resilience is related to questions of interrupting instruction or academic continuity [39] and reflects the capacity of HEIs and their academics to continue to provide teaching and learning after a disturbing event [13], as referred to in extreme situations of weather and health, particularly in the H1N1 pandemic of 2009 [40]. Finally, Dohaney et al. [13] studied how resilience is constructed in the case of disruption, as happened with COVID-19, concluding that HEIs' support, sense of community and planning and the flexibility of online teaching are critical factors in forming and preventing that resilience, but point out that face-to-face teaching should not be underestimated, because connections are crucial for the academic community to be "feeling normal again" [13] (p. 23).

From the above, the absence of resilience in pandemics is seen to have psychological effects and emotional impacts on the academic community, in situations of confinement, where anxiety, depression and stress range from moderate to serious for the participants in a study made mainly in the areas of Arts and Humanities, Social Sciences and Law [41], corroborating other studies on the impact of this disruption on university actors [3,42,43], as well as studies with samples from other contexts [44,45]. Although there is no magic formula to deal with COVID-19, it is possible to outline strategies to face up to and reduce stress, promote resilience and recovery from negative sentiments [15], and raise positivity and optimism [16]. In other words, the pandemic has clarified that it is necessary to pay more attention to the emotions and life experiences of students, Ph.D. students, colleagues and co-authors, as everyone is living with the uncertainty of COVID-19 [11]. Among other considerations, Corbera et al. [11] highlighted Ph.D. students who are halfway through their thesis and in confinement. They need to feel there is flexibility regarding deadlines, expectations and even hope. In the same line of thought, Inouye et al. [18] stated that master, Ph.D. and post-Ph.D. students should be protected by HEIs and other scientific institutions, as losing one year of data can affect their ability to complete their theses and prevent them from embarking on a career. As such, the restrictions induced by the pandemic on research, in general, are omnipresent, with these authors arguing that the difficulties of researchers obtaining data in the field and who are in the early stages of their career should be recognised. When speaking about young researchers, these will probably be the projects of Ph.D. and post-Ph.D. students, who have an essential role in many research nuclei. However, the pandemic casts doubt on the financing of their projects, with many HEIs freezing existing and new contracts, adding to the precariousness of these researchers [46].

## 2.2. Practical Effects of the COVID-19 on Academic Context

It is often highlighted that previous experiences are learning for the future, and so experimenting with previous infectious outbreaks, and the COVID-19 pandemic reminds us of the value of research and development, i.e., of scientific research [47]. However, a study by Myers et al. [12], with a sample of 4500 researchers in Europe and the USA, showed a sharp decline in the number of hours they devoted to research, despite the heterogeneity of responses according to the area of research and gender. A new variable mentioned by participants was also highlighted—family commitments—which suggests some new

approaches in order to achieve a balance between work and family [17,48]. Moreover, Myers et al. [12] reported that participants are pessimistic with regard to future publications. This pessimism results from scientific journals having slowed down peer-reviewing and publication due to COVID-19 [11] and may also be related to the major interruption in research on the ground [49]. On the other hand, this sudden stop means that "many institutions are now planning or implementing a ramp-up of on-site research activities, which offers an opportunity to begin implementing policies and practices that will lay the groundwork for the eventual re-opening of additional on-site academic programming, including teaching. To ramp up safely, institutions are working with stakeholder groups— such as public health experts, as well as faculty, staff and students—to develop guiding principles that will help inform and drive decision-making over the coming months" [49] (p. 1). This means that COVID-19 has implied new responsibilities and institutional challenges for HEIs and their research centres [48]. However, this pandemic has also represented a challenge for researchers who had to adapt to restricted access to their centres/laboratories [10,48], continuing with their research activities in the family home [48], with their communication and collaboration networks [48–50] and an opportunity to begin research into COVID-19 [48] and on the environmental effects of this global lockdown (e.g., pollution, anthropogeny) [51]. It is important to point out that the reduction and/or stagnation of scientific research has economic impacts on long-term economic development worldwide, despite this interruption varying according to the R0 rate of the spread of COVID-19 [49].

Reading of this figure leads to arguing that COVID-19 has meant that planning for resilience should be a future priority for the academic community, which "has much work to do to improve disaster resiliency" [49], as it should never be forgotten that HEIs are places of unrestricted development, open minds, sharing and transferring knowledge and ideas [10]. Given this scenario, it is argued that this pandemic environment should be a driving force for HEIs to draw the lessons required to transform their mission into something that transmits values based on resilience and the ability to overcome challenges. In other words, HEIs are the main source of knowledge and its dissemination, and future blockages should not hinder this, so it is essential that they learn essential skills and that they think increasingly out of the box. In this context, Corbera et al. [11] outlined the pillars that HEIs should adopt after the pandemic and, in this way, overcome the adverse consequences caused by COVID-19 in scientific academia (Table 1).

The main conclusion drawn from Table 1 is that all the actors involved in the academic community are important for its transformation, for it to become less vulnerable to future calamities [13] and commit to being friendly academia [52]. In short, the pillars mentioned in Table 1 are based on the axes of the wellbeing of researchers, teachers and students at any level of education and should take into consideration gender inequalities and the importance of directing funds to the emerging situations. The focus on e-learning methodology should still be visible but in a balanced way for students, teachers and researchers. Finally, social and environmental sustainability should be an objective to be increasingly achieved by HEIs, so that future disruptions are overcome by the resilience and without affecting the quality of academic and non-academic life of those involved

Based on these arguments, the following section describes the methodology used to obtain the empirical evidence on the topic studied here.

**Table 1.** Guidelines for renewed academic practice during and post crisis.

| What | How | Who |
|---|---|---|
| Tasks and priorities | Prioritise personal and collective wellbeing over "productivity" focused tasks, recognise the diversity of needs, experiences and vulnerabilities during the crisis, and question overall "rat race" practices. | Faculty, Administrators |
| Inequalities | Reflect on how the COVID-19 crisis is widening gender, ethnic and class inequalities and acknowledge them openly and collectively. Act upon inequalities in academic institutional environments through additional recognition of and funding and technical support to vulnerable groups at all academic levels. | All members of the academic community |
| Emergency support | Redirect funding originally earmarked for non-essential travel and other non-core costs to cover student, postdoctoral and adjunct faculty emergencies and other practices focusing on wellbeing and direct support to more vulnerable groups | Administrators and Faculty |
| Remote teleworking | During the crisis, organise meetings that focus on care and support in addition to "business-focused" meetings. After the crisis, increase use of teleworking and teleconferencing options when logistically feasible while respecting participants' constraints (parent care, childcare). Aim for a parsimonious and efficient academic task management and avoid the over-scheduling of teleconferences | University administrators and Faculty |
| Remote teaching | During the crisis, consider the many differences and inequalities among students and teachers in their ability to participate in remote teaching and learning, and adjust participation and evaluation criteria accordingly. After the crisis, carefully weigh in the strengths and limitations of increased remote teaching for different fields and courses | Administrators and Instructors |
| Research practices | Establish new practices for data collection and dataset sharing as well as overall collaborative research and writing. Consider and minimise environmental impacts | All Researchers |
| Dissemination | Consider moving yearly conferences and workshops to smaller online meetings every two years in order to cut on carbon emissions and allow for greater participation of low-income or/and geographically remote participants. Those online conferences/workshops could be spread throughout the year. | Faculty, Administrators, and Meeting organisers |
| Productivity | Challenge productivity measures (i.e., number of academic papers, impact factors, citation indexes and hypercompetitive funding) as the only priority evaluation criteria. Add (or push funders to add) evaluation criteria such as direct support to medical or social emergency during crises; community or policy work related to social, economic, environmental and political issues in crises; direct support to colleagues, students and other university collectives during and in the aftermath of crises | Administrators and Funders |
| Evaluation | During crisis, extend (or push to) timeline for faculty promotion, evaluation, and tenure by one year. Extend (or push to) timeline for grant eligibility or assessment criteria by one semester or one year | Administrators and Funders |
| Hiring | Prioritise (or push to) the creation of long-term academic positions over short-term, adjunct faculty members and instructors. Increase pay compensations for adjunct teaching staff, including online teachers | Administrators and Funders |

Source: Adapted from [11] (p. 6).

## 3. Methodology

### 3.1. Type of Study and Data Collection

Given the research objective defined for our study, in this study, the mixed method was adopted, consisting of both quantitative and qualitative approaches [53]. The use of a qualitative (content analysis) and quantitative (descriptive analysis) methodology allows studying this type of phenomenon [54]. These approaches are pertinent when a

phenomenon is unknown and multifaceted. On the other hand, when an emergent response from science is required, this methodology becomes useful [55], especially when we intend to study academic researchers who were deprived of their academies, laboratories and HEIs to develop their activities as a consequence of the first general confinement in March 2020. Several studies have used this method to study crisis effects [56–59], so this approach allows to extend the theory [60–62]. For Gardner et al. [63], in the last three decades, a growing number of studies have used mixed methods.

The COVID-19 lockdown showed the need to undertake empirical studies to determine its effects on researchers' academic life. In this context, the fieldwork was conducted using several data collection methods (qualitative and quantitative). Qualitative empirical data were gathered via open questions and documental analysis, while quantitative data were obtained through a questionnaire. As Yin [54] states, the adoption of various data sources is relevant, as it allows increased validity of the construct and reliability of a study.

Therefore, data collection was based on a brief questionnaire with closed and open questions. This instrument, elaborated according to the literature review, aims to determine the effects of this pandemic on research activities and others associated with academic staff. The population included lecturers, researchers and Ph.D. and master students, where aspects associated with the experience and education were criteria considered. Thus, snowball sampling (non-probabilistic) was used [64], with the questionnaire being launched on social networks [65].

The questionnaire included 10 questions, 9 with closed answers and 1 with an open answer. The questions asked covered the following topics: area of research, academic post, the HEI where they are carrying out the activity and the respective country, impacts on academic activity, time management, delays in planning, economic and operational damage and the effect on collecting data for research. The open question asked for a general opinion on the effects of COVID-19 on their academic activities. The questionnaire was subjected to a pre-test so as to validate the vocabulary used in the questions and ensure the latter allowed us to reach the intended objectives.

The questionnaire was published on social networks on 7 April 2020, and the number of answers obtained up to 8 June 2020 was 254, therefore a sample of 254 participants. This way of distributing questionnaires has been used frequently in recent times and brings new potential for experimenting [66].

*3.2. Data Analysis*

After the selection of the data-gathering instrument, we proceeded to organisation and analysis. The answers received were automatically entered on an Excel map to be able to use the mixed methodology. In other words, the descriptive analysis (tables, graphics) was applied to summarise the closed answers received and the inherent content analysis for the open question. Weber [67] and Patton [64] defined content analysis as a technique of investigation that allows objective, systematic and quantitative description of the content shown in communications, with the aim to interpret it. For the questionnaire (closed questions) used (quantitative method), the data were treated with descriptive analysis using absolute and relative frequencies and mean values.

To summarise, scientific validation was made of all the information treated through the technique called triangulation, which consists of comparing information from multiple sources of evidence (closed and open questions) so as to determine coherence, accuracy and reliability.

## 4. Results and Discussion

Aiming to obtain information about the real situation experienced by researchers regarding the effects of COVID-19 on their scientific activities and careers, the 254 answers received led to identifying some relevant aspects. The next section presents the evidence obtained summarised in short texts, figures and tables, followed by their discussion.



The participants (lecturers/researchers, master and Ph.D. students) carry out research mainly in the areas of Applied Sciences and Social Sciences, as seen in Figure 1.

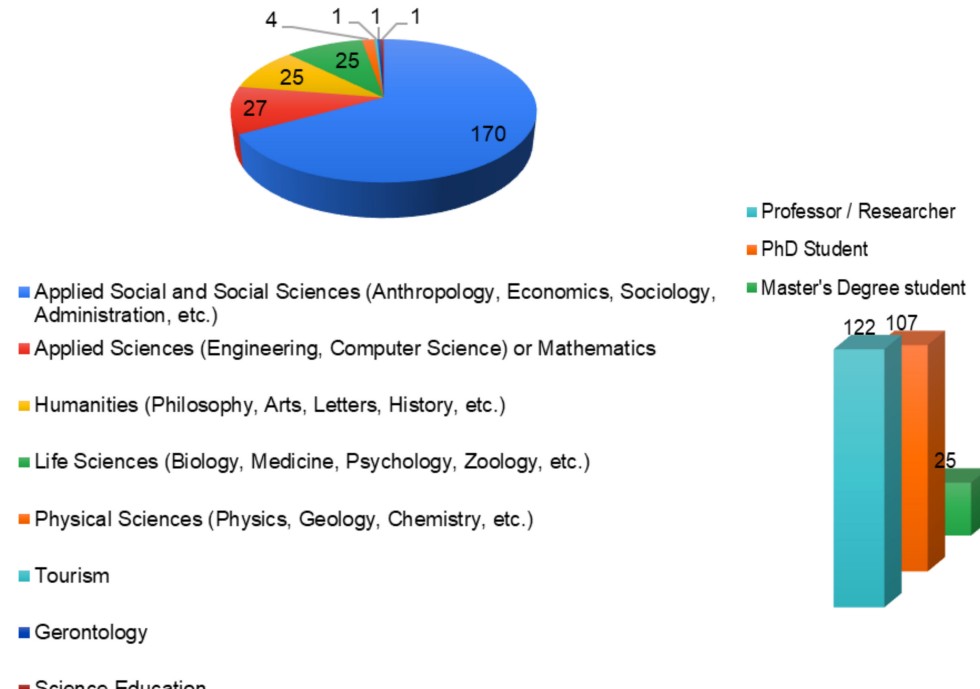

**Figure 1.** Areas of research and participants' occupation (Source: research data).

Concerning participants' country of origin, Portugal stands out with 195 answers, followed by Brazil with 38 and Angola with 7, the remaining countries having 1 or 2 responses (e.g., Lithuania, Italy, France, Spain, China, Germany). It is important to refer to China, the response being from a Ph.D. student in the field of Life Sciences at Wuhan Tech University, who commented: "in *general, I had no great problems*". As for the HEIs participants belong to, the following Portuguese universities stand out: University of Beira Interior (37), University of Trás-os-Montes and Alto Douro (28), Institute of Economics and Management (19), University of Aveiro (14), Faculty of Social and Human Sciences at the New University (13), University of Minho (13), University of Porto (13), University of Coimbra (10), Leiria Polytechnic Institute (10), in the research areas of Social Sciences and Applied Social Sciences.

After the generic characterisation of the sample, Figure 2 presents a summary of the answer to the question: *In general, to what extent has your work been affected by the COVID-19 epidemic?*

Analysis of Figure 3 reveals that for most participants, the measures imposed by COVID-19 had a severe impact (5—extremely affected) on their academic activities. Plausible explanations will have been transitioned to the non-face-to-face system in all those activities, with rapid adaptation being necessary to create a working environment at home, together with a new family routine. Table 2 divides the general impact (Likert Scale (1 to 5—no impact to high impact)) according to areas of research and occupation.

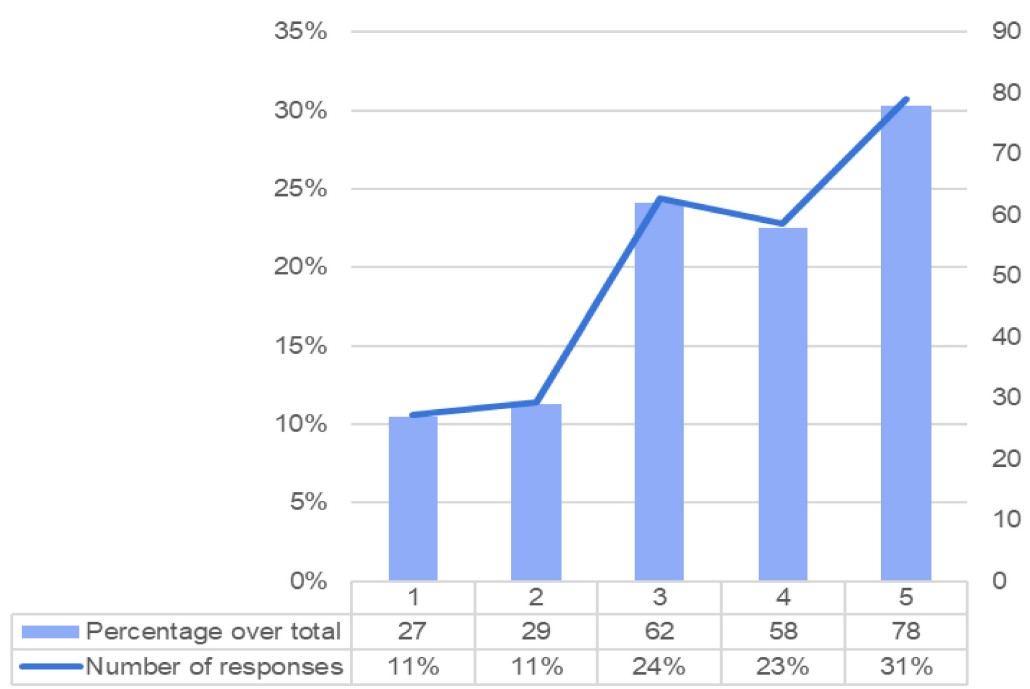

**Figure 2.** Likert scale (Source: research data).

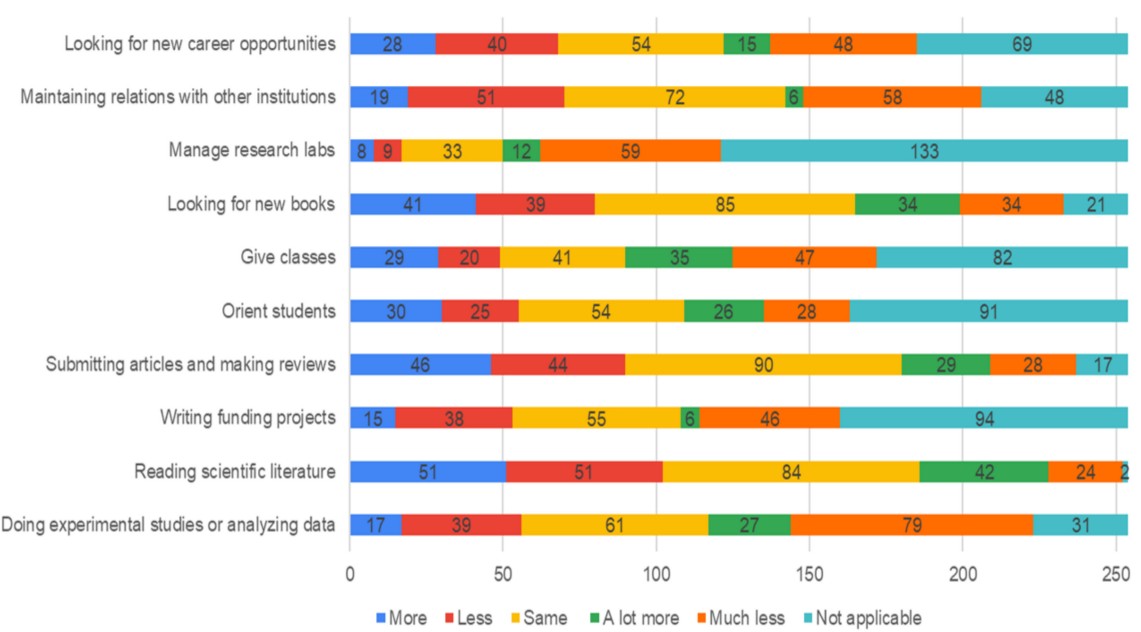

**Figure 3.** Typology of time management in the various academic activities (Source: Research data).

Table 2. General impact on academic activities.

| Research Área/Impact Level | Teacher/Researcher | | | | | Doctoral and Master's Students | | | | |
|---|---|---|---|---|---|---|---|---|---|---|
| | 1 | 2 | 3 | 4 | 5 | 1 | 2 | 3 | 4 | 5 |
| Applied Social and Social Sciences (Anthropology, Economics, Sociology, Administration, etc.) | 12 | 11 | 22 | 19 | 33 | 4 | 2 | 20 | 20 | 18 |
| Applied Sciences (Engineering, Computer Science) or Mathematics | 2 | | 6 | 3 | | 3 | 10 | 2 | 3 | 6 |
| Humanities (Philosophy, Arts, Letters, History, etc.) | 1 | | 1 | 1 | 2 | 1 | 4 | 6 | 4 | 5 |
| Life Sciences (Biology, Medicine, Psychology, Zoology, etc.) | | | 2 | 1 | 3 | 3 | 2 | 2 | 6 | 7 |
| Physical Sciences (Physics, Geology, Chemistry, etc.) | | 1 | | | 1 | 1 | 1 | | | 2 |
| Tourism | | | | | | 1 | | | | |
| Gerontology | | | | | | | | | 1 | |
| Science Education | | | | 1 | | | | | | |
| **Total** | **15** | **12** | **32** | **24** | **40** | **12** | **17** | **30** | **34** | **38** |

Source: research data.

Table 2 shows a minimal difference in the impact of COVID-19 on lecturers/researchers and Ph.D. and master students, 48.4% (123 participants) and 51.6% (131 participants), respectively. For Dohaney et al. (2020), pandemics create disruption in science in academia, with it being crucial for HEIs to define and implement contingency plans. The serious impact found here may mean these plans do not exist since it was the academics themselves who had to draw up their plan B to adapt to a new form of teaching and research at home, overcoming negative feelings and the challenges this created [30]. This means that some HEIs were not so quick to provide a response of continuity, as claimed by Carver [30].

Although the impact of HEIs' closure was severe for the majority of respondents, this was crucial to reduce the spread of infectious diseases and break transmission chains [4–6]. From another perspective, it would be important for the academic actors analysed here to face this disruption as an opportunity and a challenge to reconsider the educational system since the return to new normality implies re-learning to operate, function and communicate [11].

Another question asked in this study was related to time management in the various academic activities: Are you spending more, less or the same amount of time on the following activities, compared with the time you spent before the COVID-19 epidemic? Figure 3 and Table 3 present the answers to this question, where descriptive statistics were used.

Table 3. Descriptive statistics of Q6: Time management in performing activities.

| Questions | Doing Experimental Studies or Analysing Data | Reading Scientific Literature | Writing Funding Projects | Submitting Articles and Making Reviews | Orient Students | Give Classes | Looking for New Books | Manage Research Labs | Maintaining Relations with Other Institutions | Looking for New Career Opportunities |
|---|---|---|---|---|---|---|---|---|---|---|
| Mean | 2.64 | 2.99 | 3.31 | 3.04 | 3.62 | 3.45 | 3.04 | 3.63 | 2.82 | 3.18 |

Source: research data.

Figure 4 shows that academics consider the various hypotheses presented in this question are not applicable in their daily time management or consider that they spend the same number of hours on these activities. As a complement, Table 3 presents the descriptive statistics for this question.

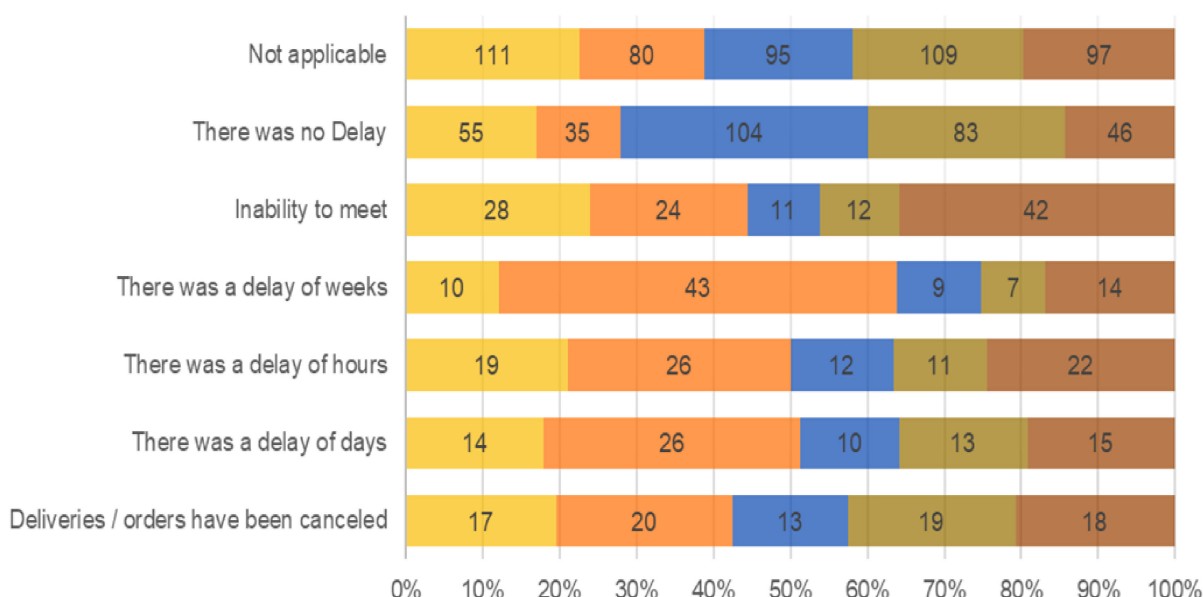

**Figure 4.** Characterisation of economic and operational damage (Source: research data).

This question presented various response options, with a score to be given between a minimum and maximum value. It was related to time management in carrying out academic activities at a time of COVID-19. Afterward, 10 response options were presented to be scored from "More", "Less", "Same", "A lot more", "Much less" and "Not applicable". To code these answers, values from 1 to 6 were attributed, according to the order of the response options. In the ten response options, the 254 participants gave an average score between 2 and 4, meaning that they fell within the response options of "Less", "Same" and "A lot more". Therefore, COVID-19 is seen to have various types of influence on researchers' time management of their scientific activities. The questions with the highest average scores were "Orient students", "Give classes" and "Manage research labs", with respondents saying that the impact of COVID-19 hindered their time management in performing these activities. The items with the lowest scores were "Doing experimental studies or analyzing data", "Reading scientific literature" and "Maintaining relations with other institutions", where the respondents revealed that the impact of COVID-19 was irrelevant, as time management for these activities was unchanged.

Although the 254 participants reported that COVID-19 had a relevant impact on their academic activities, when asked about changes in time management of these activities, the answers received showed a positive response. This means that respondents accepted the challenge suddenly presented by the restrictions, as claimed by various authors [10,48–50].

The answers to the question: What impact did the epidemic have on timetabling of your main research project? Are summarised in Table 4.

Research timetables are seen to be affected considerably for most respondents (193), representing 76% of the sample, with Ph.D. and master students representing 53% and lecturers/researchers 47%. Although these percentages are similar, it is fundamental for students to adhere to schedules in producing their theses, particularly at the data-collection stage, due to the rigour demanded. Obviously, lecturers/researchers' timetables are also crucial for their research, for example, participation in conferences that were

cancelled, international mobility and re-allocation of resources, among other factors. This means that the confinement imposed by COVID-19 changed considerably the schedules defined and implemented for completing their current research due to the impossibility of accessing physical spaces, gathering primary data, meeting up with teams face-to-face and many other activities. The typology of responses indicates that researchers, whatever their academic status, should be protected by HEIs and other institutions, as defended by Ahmed et al. [46] and Inouye et al. [18], who stated that when researchers lose data and financing, this causes irreparable damage to their academic careers.

**Table 4.** Impacts on research schedules.

| Description | Lecturers/Researchers | Ph.D and Master Students | Responses |
|:---:|:---:|:---:|:---:|
| Great impact | 42 | 39 | 81 |
| Average impact | 28 | 28 | 56 |
| High impact | 21 | 35 | 56 |
| No impact | 19 | 18 | 37 |
| Minimum impact | 11 | 13 | 24 |
| **Total** | **121** | **133** | **254** |

Source: research data.

To assess the level of harm to scientific activities, the following question was asked: *How much economic or operational harm was caused to your Department or Laboratory or Start-up by the epidemic?* To characterise that harm, various options were proposed associated with five areas that could be affected, as shown in Figure 5.

| Personal barriers | |
|:---:|:---|
| 1 | Various negative and psychological impacts |
| 2 | Distance / isolation from family and friends |
| 3 | No interaction and connections |
| 4 | More family assistance and increased housework |
| 5 | Postponement of classes |
| 6 | Quick adaptation to the online format |
| 7 | Longer workload |
| 8 | Loss of productivity |
| 9 | HEIs do not have a plan B |

| Professional barriers | |
|:---:|:---|
| 1 | Difficulty obtaining answers to questionnaires |
| 2 | Complete change of the thesis, as the data lost its validity |
| 3 | Logistical difficulties |
| 4 | Difficulty in data collection |
| 5 | Impossibility of carrying out fieldwork |
| 6 | Inability to access laboratories |
| 7 | Failure to meet deadlines |
| 8 | Continuity issues |
| 9 | Cuts in research grants and funding in some areas |

| Opportunities | |
|:---:|:---|
| 1 | Positive impact of teleworking |
| 2 | Up-to-date work placement |
| 3 | Increased efficiency in time management with teleworking |

| Opportunities for the near future | |
|:---:|:---|
| 1 | Extension of deadlines for submission of theses by HEIs |
| 2 | Extension of project deadlines |
| 3 | Importance of more financial allocation |

**Figure 5.** Summary of opinions on the impact of COVID-19 (Source: research data).

Figure 4 shows that administrative management of physical spaces and the resources associated with academic activities gave an average score of between 4 and 5, corresponding to "There was a delay of weeks" and "Inability to meet". Table 5 revealed that the impact of COVID-19 on economic and operational harm related to research activities was characterized by direct impacts on research laboratories, service provision by suppliers,

delivery of products and/or services to clients, capacity to pay salaries, receiving funding to maintain laboratory infrastructure and even collaborators' ability to carry out their normal work. The respondents indicated that in relation to the typology of the above impacts, delays were of weeks or more, in some cases resulting in incalculable amounts.

**Table 5.** Descriptive statistics of the characterisation of economic and operational harm (absolute frequency).

| Questions | Delivery of Services, Materials and Suppliers by Suppliers of the Lab or Company | Completion and Every of Research Tasks, or Delivery of Products and Services to Customers | Salary of Grantees /Employees | Receipt of Funds for Maintenance of the Lab, or Receipt of Payments for Customers | Capacity of the Grantees/Collaborators to Come Daily to the Lab |
|---|---|---|---|---|---|
| Mean | 4.70 | 4.48 | 3.91 | 4.37 | 4.68 |

Source: research data.

In pandemics, it is important to continue to ensure management of physical spaces and the associated resources, which was achieved by most of the respondents, and when analysed holistically with the previous questions, reveals the resilience of these academics, as mentioned by several authors [14,32,33,41].

The results presented here show that the impacts of COVID-19 in the academic sphere resulted from rising to meet a challenge by ensuring the continuity of academic activities resorting to technological tools, which does not mean that severe impacts were not felt in some activities.

These results find some support in the scarce literature on the topic, particularly that academics were scientifically responsive by prioritising collective objectives over individual ones [11]. However, this does not mean that the disruption coped with did not cause psychological effects on those involved, as reported in previous research on other public health crises [34–36]. Confinement and social isolation, feelings of fear and frustration are predictors of psychological changes in people, which can later be transformed into stress and anxiety [33]. It is precisely how these feelings are combated that makes all the difference, i.e., resilience is central to people's continued mental health [14].

Therefore the academics studied here are equipped with educational and institutional resilience [13]. On the other hand, the 132 Ph.D. and master students included in this analysis demonstrated little educational resilience in the adverse circumstances (51.6% answered that COVID-19 had serious impacts on their scientific activities). As Vance et al. (2015) explained, for there to be educational resilience, students must be able to cope with and prosper from adverse events. Lecturers/researchers must also be resilient to overcome the consequences of the COVID-19 pandemic, i.e., they must have the capacity to exercise institutional resilience [39]. However, 48.4% of them highlighted that their scientific activities were severely affected. This low institutional resilience may be related to HEIs' delayed response in changing from face-to-face to online teaching since this is defined according to HEIs' response capacity in unexpected situations [13].

From another perspective, the question of resilience is more evident in Applied Social and Social Sciences (Anthropology, Economics, Sociology, Administration, etc.), which corroborates other studies [3,41–43]. However, if we compare these degrees of resilience with how these academics manage their time, physical research premises and associated resources, their institutional resilience is considerably higher. Finally, the restrictions and issues caused by HEIs' closure inevitably led to relevant changes in research schedules, casting doubt on the future of many scientific investigations in progress, and so these academics should be supported by their host institutions, as proposed by Inouye et al. [18], for example.

To determine how the COVID-19 context influenced data collection and inherent procedures, the 254 participants answered the following questions: *What is your source of data? How did you obtain data for your research?* Table 6 shows the results obtained.

**Table 6.** Sources and collection of data for research.

| Description | Answers |
| --- | --- |
| I am responsible for collecting, processing and analysing the data | 147 |
| The research is entirely bibliographic, and I only analyse the available data | 19 |
| I am responsible for collecting the data, which was processed by someone else, and analysed by me | 11 |
| The data are secondary, obtained from another institution/author, but were processed and analysed by me | 5 |
| I am responsible for collecting, processing and analysing the data. The research is entirely bibliographic, and I only perform the analysis of the available data. | 5 |
| The data were collected by someone else on my team, but was processed and analysed by me | 4 |
| This question does not apply to all contexts so narrowly. | 4 |
| The data were collected by someone else on my team, and was also processed by someone else, but only analysed by me | 2 |
| Interviews conducted and treated by student | 2 |
| Other answers | 55 |
| **Total** | **254** |

Source: Research data.

Analysis of Table 6 reveals that the data sources used are primary and secondary, that most researchers are responsible for the whole process of data collection, treatment and analysis (147), that many operate as a group (research teams, 46 researchers), that four researchers considered the question should not be so restricted, as it does not apply to all research environments, and that two researchers asked students to carry out data collection and treatment. In addition, 19 researchers carry out bibliographical research and only analyse available data.

As for other answers (55), these are extremely heterogeneous and cannot be grouped, for example, "for data treatment and analysis I depend on others", "we are not yet collecting data", "data are being collected by students", "we are at a preparatory stage". Therefore, apparently, COVID -19 did not change the way of obtaining and treating data. This may only have been put off till later, since interviews could no longer be held, data collection on the ground was not possible and people's availability to answer online questionnaires may have been limited due to social isolation.

Finally, the open question: *Would you like to say anything about the impact of the COVID-19 epidemic on your research activity?* Here, it is noted that 134 researchers chose not to answer. However, in the 120 answers obtained, the words "*anxiety, emotional, stress, psychological, fear, concentration, teleworking, tasks, online*" often appeared. In other words, psychological factors and rapid adaptation to the online format were a constant for these participants, which tested their resilience, particularly that of women who might also have to care for their families (e.g., children). Another analysis made of these 120 answers allowed identification of the barriers, successes obtained and opportunities for the future (Figure 6).

Analysis of Figure 5, among other important questions, shows the importance of open questions, i.e., qualitative methodology, to obtain results about a social phenomenon. Clearly, the answers given were not visible in the closed questions discussed above. The answers showed that the academic actors had to overcome their own barriers to face the social and psychological effects of living with a pandemic. A new variable emerges in this answer, the increased number of hours dedicated to the family, mainly children, which corroborates recent research [12,17,48]. According to the literature reviewed, these barriers can be overcome by adopting strategies to promote resilience, such as optimism and positivity [15,16], and academia should be alert to its actors' emotions and life experiences [11].

Furthermore, these academics faced professional barriers connected to data collection, logistic restrictions and funding for research, among others, and so Inouye et al. [18] considered it important to recognise the difficulties they face, and Corbera et al. [11] concluded they need to feel there was flexibility regarding deadlines, expectations and even hope. However, some academics included in this study referred to some present and future opportunities created by COVID-19. Specifically, some considered that quarantine led to more efficient time management through teleworking. This should not mean replacing face-to-face with online teaching, as the former is important in the educational context [13]. For the future, the extension of deadlines was suggested, more financing for research to mitigate the consequences of COVID-19 for research schedules.

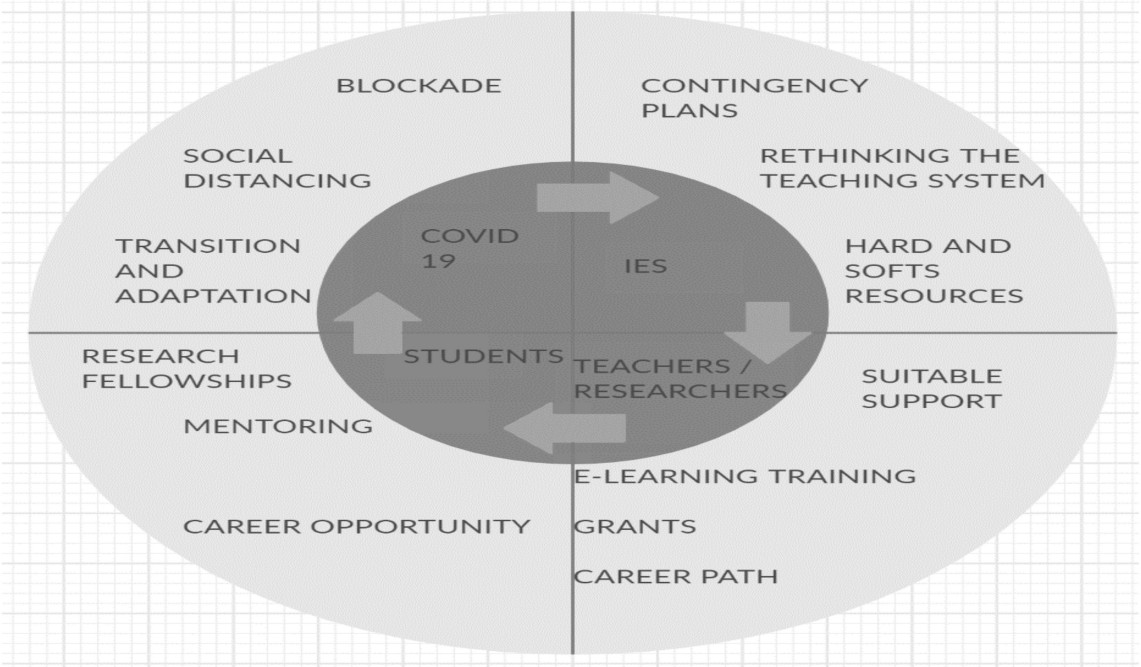

**Figure 6.** Theoretical framework of a new Plan B for HEIs (Source: From research data).

This study revealed academics' state of mind regarding the effects of COVID-19, the results suggesting some present and future responsibilities and challenges for HEIs and all their actors [11,48], so that future calamities do not cause the disruption portrayed by Corbera et al. [11]. In this context, Figure 6 shows a theoretical framework for HEIs' institutional renewal, based on the harsh experiences provoked by the current pandemic.

This framework emphasises the importance of returning to a normality that reflects the feeling of an academic community in all its vectors and the joint involvement of all in order to achieve a fundamental common aim, the advance of scientific knowledge [47]. Moreover, this framework is in line with the following argument: *"The notion of an educator as a knowledge holder who imparts wisdom to his or her students is no longer adequate for the purpose of 21st century education. With students being able to access knowledge, and even learn a technical skill, through a few clicks on their phones, tablets and computers, we will have to redefine the role of the educator in the classroom and lecture theatre. This may mean that the role of educators will have to move towards facilitating the development of young people as contributing members of society"* [65], p. 1, that is, the future requires academics to be increasingly resilient, to engage in continuous learning, to be flexible to adapt quickly to disruptions and to address these with their entrepreneurial skills. It is also important that academics acquire emotional intelligence skills to overcome future crises with creativity, critical and holistic thinking, and, in this way, minimise the negative effects of blockages in the sharing and transfer of knowledge by HEIs.

## 5. Conclusions and Contributions

Declaring the worldwide pandemic drew greater attention to the importance of scientific research, not only in Life Sciences but in all areas of knowledge, and also the role played by academics of any type in developing theoretical and empirical research on topics still in need of this. COVID-19 is one of those topics where there is clearly a lack of experience and knowledge of its effects in all scientific domains. Therefore, the main contribution of this empirical study lies in gaining the perception of lecturers, researchers and Ph.D. and master students regarding the effects of COVID-19 on their various academic activities. The descriptive and content analysis resulted in identifying the barriers and opportunities experienced during the global stoppage of all economic and social activity.

The theoretical implications are reflected in the theoretical framework presented, emphasising an integrated and holistic academic community of all its actors in order to face the future with educational and institutional resilience, which was recommended as a potential strategy to cope with adverse and unavoidable situations. Moreover, this model intends HEIs to continue to be a place of open, innovative minds, where all ideas and projects are valued and supported, with opportunities for all, with a feeling of commitment to the common good, and as stated by Cordeza et al. [11], with scientific answers for the future which does not appear easy, but rather full of uncertainties. Another contribution lies in obtaining empirical evidence about the effects of this virus, for which there is still some shortage.

From the results obtained, it stands out that most answers revealed severe impacts on academic activities but not related to the management of the time allocated to each task or with a harmful effect on the operational management of spaces. This means that those impacts will possibly be more associated with these tasks having to be performed at home, surrounded by the family, without much knowledge of online teaching and what tools to use (there was no time for training), which required a sudden transition and adaptation to prevent the disruption having an even greater effect on institutional resilience.

As implications for practice, the evidence points to the need for academics to be provided with training in E-learning, about technological tools for use in distance-learning, and to reconsider how they carry out their research on the ground, among others.

Like any study, this one is not without limitations. The first is that respondents are mainly from the areas of Social Sciences and Applied Sciences. While not taking away from the importance of these areas and others, the study would have been more complete if, for example, Life Sciences had been similarly represented in the sample, allowing a comparison to be made between areas and giving more added value to the study. This limitation suggests a future line of research.

The second limitation concerns the snowball-type sampling, which, although widely used, may not give the intended result. Therefore, another suggestion would be to apply the questionnaire in HEIs and their research centres, for example, in the Portuguese case, one per district. The third limitation was the added difficulty of carrying out a search for articles in the main online databases to acquire pertinent literature for the literature review, as there is a great number of non-peer-reviewed publications and studies by European entities (e.g., OCED, UNESCO, World Economic Forum). To remedy this limitation, it is relevant and urgent for researchers to resume this academic activity so as to advance knowledge to be able to cope better with and overcome future pandemics and other calamities, and also for scientific journals to resume peer-review processes for articles submitted on these topics. The final limitation results from participants being mainly from Portugal, suggesting replication in other geographical contexts through international social networks.

Another limitation is related to the sample size, marked by the low representation of other areas of knowledge since the area of Social Sciences and Humanities accounted for most of the answers. Thus, future studies should address this limitation, as it is different to investigate in laboratories with data collected in the field and research with data obtained by other means. Moreover, the fact that most of the answers were from academics from

Portuguese HEIs is a further limitation, so it would be interesting, in the future, to analyse only these answers and fit them into the higher education model in Portugal.

Finally, researching and elaborating a study on any topic related to COVID-19 is always a challenge for any researcher, given the lack of data and supporting literature. In these circumstances, resilience and persistence were crucial in defining this quasi-experimental empirical study through snowball sampling and, therefore, with no forecast of the sample size, which is so important for the credibility and scientificity of any research. Nevertheless, a sample of a relevant size was obtained, which allowed forming a series of constructs on the effects of this virus on academic activities. The constructs to retain are the psychological and emotional impacts provoked by COVID-19, which, if not faced with resilience by all academics, would have negative personal and institutional consequences: the urgency of reconsidering the teaching system to put into practice in the near future, without putting health at risk, but also without neglecting face-to-face teaching, so it is important to develop the feeling of belonging to the academic community; the importance of using technological tools even in face-to-face classes, access to and the support of technological resources for all, to promote equality and social cohesion; retaining that scientific research is a driver of good practices in all areas of knowledge and added value in any situation, and as such, it is crucial that research does not stagnate in fields such as management and business.

In the certainty that subsequent studies will appear, it is hoped that the present one has contributed to stimulating more research on this and other topics associated with COVID-19 so that contingency plans are in place when facing future calamities, rather than having to learn immediately on the spot. HEIs that acquire digital resources and a possible system of rotational mixed teaching at the present time (face to face and e-learning) will gain resilience to face the post-COVID-19 and future adverse events.

**Author Contributions:** Formal analysis, R.S.; investigation, M.R.; supervision, M.F.; writing—original draft M.R.; writing—review and editing, M.R., M.F. and R.S. All authors have read and agreed to the published version of the manuscript.

**Funding:** This research received no external funding.

**Institutional Review Board Statement:** Not applicable.

**Informed Consent Statement:** Not applicable.

**Data Availability Statement:** Not applicable.

**Acknowledgments:** The authors are grateful to the anonymous referees of the journal for their extremely useful suggestions to improve the quality of the paper. The authors gratefully acknowledge financial support from the National Funds of the FCT—Portuguese Foundation for Science and Technology within the project «UIDB/04007/2020» and the project «UIDB/04011/2020».

**Conflicts of Interest:** The authors declare no conflict of interest.

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
