# Peer review of "Teaching and Researching in the Context of COVID-19: An Empirical Study in Higher Education"

_sustainability, doi:10.3390/su13168718_

Round 1

Reviewer 1 Report

The study is interesting and makes relevant contributions. However, there are certain aspects that must be improved:

  1. ABSTRACT:

1.1 It would be interesting to raise several specific implications, starting from the identification of barriers and opportunities, as well as including them at the end of the abstract. What is proposed in the last lines (21-23) is too general. There are several implications included in the “Final comments and contributions section”. Therefore, they could be retrieved and presented in the abstract.

  1. LITERATURE REVIEW

2.1 It would be advisable to divide the information in the “Literature Review” section, considering the aim of the research.

2.2 It does not make much sense to present twice the information that is included in Figure 1 and Table 1. It is better to leave the table.

  1. METHODOLOGY

3.1 It is evident that there is some confusion with the methodological prism that has been chosen to develop the research. This is a fundamental question. In fact, the methodology section, which we could consider the most important, is the weakest.

3.2 If we look at the abstract, it is stated that the quantitative research methodology has been chosen. However, at the end of the introduction it is stated that a mixed methodology is used, when in the method section it is indicated that the methodology that has been selected is the qualitative one. Considering the data, it can be seen that the approach is mixed, for which a broad foundation would have to be provided to justify, rigorously, this choice.

3.3 It would be convenient to justify each one of the decisions that were taken in the research process. We cannot say “the population is not exact…” or “may be the only way…”.

3.4 Information of great interest is obviated. Questions of the type arise:

- What characteristics does the sample have?

- Why was social networks chosen as a means of dissemination of the instrument? What risks does this option entail?

- Was the questionnaire validated?

- What about the analysis procedure? What kind of analysis is done at a quantitative level? How is content analysis developed? What type of content analysis is done and why? Is any instrument used to facilitate this process?

  1. RESULTS

4.1 Personally, I do not present results and discussion in the same section. Maybe, for this decision, discussion is too short in this article.

4.2 The conclusion reached after Figure 1 is somewhat relative because we do not know what specialties  the participants are dedicated to (we discover this later in Table 2). Most belonged to the field of Social Sciences, hence the predominance of these investigations.

4.3. I would recommend using the APA format for the presentation of tables and figures. It is also advisable to unify criteria with the font size in the different tables. In Table 5, for example, the text is not easy to read.

4.4 I understand the quantitative data that appears in Figure 6 to be the absolute frequency. It should be indicated and explained what it is.

  1. OTHER QUESTIONS

5.1 It is recommended not to use quotes and italics at the same time, as it is a redundancy.

Author Response

all suggestions were considered to add value to the article and were integrated in the article (red colour).

Reviewer 2 Report

The paper has a actually theme, which can help to adapt those involved in this field, both as practice and as its study, in order to improve it.
I observ that not all the figures and tables in the paper have mentions on the source, which may require reference to this aspect as well. I notice that a figure and a table  are taked integrally from reference 11, but we believe that an interpretation of the authors on the data taken in this way would have been more appropriate, in order to emphasize the originality of their work. We recommend the authors to detail through comments on each figure or table presented, the comments added after them are short, which is why I suggest to be developed. Table 2 does not explain the significance of columns marked 1-5. Line 318 refers to Figure 5, not to the table. Figure 7 requires details about the content and how it is represented.
The paper does not fully comply with the format required by the journal, to be structured on the following sections: Introduction, Materials & Methods, Results, Conclusions.
I recommend a major revision of the paper in order to publish it.

Author Response

(The authors gave the same response as above.)

Round 2

Reviewer 1 Report

I appreciate that some interesting changes have been done but I do not think it is enough: 

  1. The problem with the methodology in the abstract has not been solved. 
  2. It has not been considered the suggestion to divide the literature review section in some parts to organise better the information.
  3. The justification for the selection of the methodology is too short.
  4. Problems detected with methodology have not been solved. It would be convenient to justify each one of the decisions that were taken in the research process.
  5. I do not understand the information included in the parenthesis of Table 5. I think this data does not correspond there.
  6. The recommendations referred to the discussion have not been considered. 

Author Response

See the responses, please

Reviewer 2 Report

The paper shows improvements, except for the fact that most of the figures and tables used are not yet specified the sources that were the basis for their realization (including if they were made by the authors). The recommendation to detail through more extensive comments each figure and table used, is not integrally do it, but I respect the authors' option to present their material and research results in this paper.

Author Response

See the responses, please.
